# Exploring Genetic Diversity and Population Structure of Australian Passion Fruit Germplasm

**DOI:** 10.3390/biotech14020037

**Published:** 2025-05-16

**Authors:** Xinhang Sun, Peter Bundock, Patrick Mason, Pragya Dhakal Poudel, Rajeev Varshney, Bruce Topp, Mobashwer Alam

**Affiliations:** 1Queensland Alliance for Agriculture & Food Innovation, The University of Queensland, Nambour, QLD 4560, Australiap.dhakalpoudel@uq.edu.au (P.D.P.);; 2Southern Cross Plant Science, Southern Cross University, Lismore, NSW 2480, Australia; 3WA State Agricultural Biotechnology Centre, Centre for Crop & Food Innovation, Murdoch University, Perth, WA 6150, Australia

**Keywords:** passion fruit, *Passiflora* spp., germplasm, population structure, genetic diversity

## Abstract

Evaluating the genetic variability of germplasms is essential for enhancing and developing superior cultivars. However, there is limited information on cultivated germplasm diversity for Australian passion fruit breeding programs. The genetic diversity of Australian passion fruit (*Passiflora* spp.), including 94 rootstocks and 95 scions, was evaluated to support breeding programs aimed at enhancing productivity, fruit quality, and overall crop resilience. Rootstocks were genotyped using high-density 24k Diversity Arrays Technology (DArT)-based single-nucleotide polymorphism (SNP) markers, while genetic characterization of scions was conducted using eight simple sequence repeat (SSR) markers. The resulting genetic relationships revealed significant variation within rootstock populations. Bayesian cluster analysis in STRUCTURE showed that the rootstock population was divided into six distinct genetic groups, whereas only two subpopulations were identified among the scion accessions. SNP-based genotyping further highlighted the allelic diversity of Australian rootstocks, suggesting a rich reservoir of genetic traits for rootstock improvement. These findings underscore the importance of preserving and utilizing genetic diversity in Australian passion fruit germplasm to drive the development of superior cultivars with enhanced adaptability and performance under diverse environmental conditions.

## 1. Introduction

Passion fruit (*Passiflora* spp.) is a highly valued tropical and subtropical crop known for its distinct flavour, aroma, and richness in vitamins, minerals, and antioxidants, as well as notable medicinal properties [1]. It has gained significant global popularity in both domestic and export markets [2]. Brazil is the world’s largest producer and consumer of passion fruit, followed by Colombia, Indonesia, and Australia [3]. In Australia, passion fruit has significant commercial potential due to the increasing demand for both fresh fruit and processed products. With a total cultivation area of approximately 280 hectares, the Australian passion fruit industry produced 4711 tonnes of fruit in 2022–2023 [4], contributing approximately 29.6 million AUD annually. Despite its economic importance and diverse potential uses, the industry faces challenges including decreased fruit production and lack of improved varieties [5]. The decline in Australian passion fruit production, compared to 5196 tonnes in 2017, is largely attributed to increasing biotic and abiotic stresses [5]. One major challenge is *Fusarium* wilt, a significant disease caused by soil-borne pathogens that adversely affects passion fruit production [6]. Another serious disease is passion fruit woodiness virus, which can result in terminal necrosis, severe stunting, and total commercial loss in the affected plants [7]. Currently, no effective control method exists apart from the use of disease-resistant genotypes [8]. Additionally, the limited cold tolerance of commercial passion fruit cultivars restricts production and further increases their vulnerability to these diseases [9]. Rootstocks play a crucial role in plant health and vigour [10]. They are responsible for absorbing water and nutrients from the soil, providing resistance to diseases caused by soil-borne pathogens, and enhancing tolerance to environmental challenges [11]. Furthermore, the interactions and compatibility between rootstock and scion are essential for plant performance and productivity. Although one study suggested that rootstock and scion should be closely related in order to graft successfully [12], further research on graft compatibility of scions and diverse rootstock genotypes is needed to support future breeding programs. In this context, identifying genetic diversity in rootstock and scions will be instrumental in guiding future breeding research and improving the resilience of the Australian passion fruit industry.

Characterising genetic variability and elucidating the population structure provide valuable insights into understanding, conserving, and utilizing genetic resources [13]. This, in turn, enhances the effectiveness of breeding programs by enabling the selection of superior genotypes for crop improvement [14,15]. Passion fruit diversity has traditionally been assessed using morphological traits like flower colour and agronomic traits such as fruit weight, size and stress tolerance in affected plants [2]. However, these traits are influenced by environmental factors and developmental stages, which can limit their accuracy in reflecting genetic relationships [16]. DNA-based molecular markers provide a more reliable method for evaluating genetic diversity, supporting breeding, and ensuring the effective management and conservation of passion fruit germplasm [17]. Previous studies have assessed the genetic diversity of passion fruit using various methods, including the following markers: SSR [13,18,19,20,21], inter-simple sequence repeats (ISSR) [22,23,24], random amplified polymorphic DNA (RAPD) [14,24,25,26], and SNP [27,28] markers. Despite the significance of rootstocks in passion fruit production, limited research has explored the genetic diversity and population structure of passion fruit rootstocks in Australia. Conducting a genetic diversity study on Australian rootstock germplasm is essential for identifying core accessions for further physiological and agronomic evaluations.

Plant breeding relies on selecting diverse germplasms with desirable traits to develop improved cultivars [29]. Genome-wide markers are instrumental in uncovering genetic diversity within germplasm [30]. One of the most effective high-throughput genotyping methods is Diversity Arrays Technology sequencing (DArTseq), which enables the efficient discovery of genetic markers [31]. DArT markers have been extensively employed across various crop species such as sorghum, pea, macadamia, strawberry, oilseeds, groundnut, and common bean to facilitate genetic analyses and enhance the understanding of genetic diversity [32,33,34,35,36,37,38]. SSR molecular marker technology, on the other hand, relies on the amplification of specific DNA fragments followed by product detection through electrophoresis [39]. These markers are highly stable and reproducible, exhibit abundant polymorphism, and require minimal DNA input [40]. This study aims to investigate the genetic diversity and population structure of Australian passion fruit rootstock germplasm using a DArT SNP marker platform. In addition, the genetic diversity and population structure of Australian passion fruit scion germplasm were also investigated with SSR markers to support future study about rootstock–scion interactions. All these will offer valuable insights into preserving genetic resources, enhancing crop improvement, and optimising resource exploitation within national breeding programs.

## 2. Materials and Methods

### 2.1. Plant Materials

For the genetic diversity and population structure analysis, a total of 94 passion fruit rootstock accessions were obtained from the Australian National Passionfruit Breeding Program (PF19000). This population comprised of 15 Pandora accessions, 8 McGuffies Red accessions, 7 *Passiflora edulis flavicarpa* accessions, 53 of their hybrid accessions derived from these parental lines, 3 Heuston accessions, 7 SP16 (Sweetheart × Pandora) progeny accessions and 1 McLeod rootstock accession (Appendix A). Additionally, 95 scion accessions were collected from the same breeding program (Appendix A).

### 2.2. DNA Extraction

Fresh leaves from each accession were collected in 2018 from the PF19000 trial site in Duranbah, NSW, and placed into sterile 15 mL Falcon tubes, which were kept on dry ice. The samples were then transported to the laboratory, where they were rapidly frozen in liquid nitrogen and transferred to a −80 °C freezer for temporary storage. The DNA was extracted from around 500 mg of leaf samples from each of the accessions following the CTAB protocol described by Doyle [41]. DNA integrity was assessed by electrophoresis on a 0.8% agarose gel, and DNA concentration was measured using a spectrophotometer (Thermo Fisher Scientific, Waltham, MA, USA) [15]. The DNA was adjusted to a final concentration of 50 ng/μL and stored at 4 °C for subsequent use [2].

### 2.3. Genotyping and Data Processing

Ninety-four rootstock accessions were genotyped utilizing a high-density DArT SNP marker platform. Complexity reduction using PstI restriction enzymes was performed following the method by Wenzl et al. [42]. The HiSeq2000 sequencing platform (Illumina, San Diego, CA, USA) was implemented using Diversity Arrays as described by Barilli et al. [43]. Data cleaning was performed to eliminate low-quality and non-polymorphic markers [44]. Called variant filtering was performed using VCFtools version 0.1.16 [27]. Ninety-five scion accessions were genotyped using eight SSR markers (Appendix A). The SSR markers were selected from a database of previously utilized SSRs based on their high polymorphic content and extensive coverage of the passion fruit genome [45]. The PCR amplification reaction was performed in a 15 μL reaction volume containing 30 ng of genomic DNA, 0.1 μM of each primer, 1 U of Taq DNA polymerase, 0.2 mM dNTPs, 1.5 mM MgCl_2_, and 1× PCR buffer. The thermal cycling conditions included an initial denaturation at 94 °C for 4 min, followed by 35 cycles of denaturation at 94 °C for 1 min, annealing at 56–60 °C for 1 min, and extension at 72 °C for 3 min, with a final extension at 72 °C for 7 min. The amplified fragments were separated using 6% polyacrylamide gel electrophoresis in TAE buffer (Tris-base, acetic acid, and EDTA) at 90 V for approximately 3.5 h. The gel was stained with ethidium bromide, and DNA fragments were visualized under UV light.

### 2.4. Marker Quality and Genetic Diversity Analysis

Expected heterozygosity (*He*), a key measure of genetic diversity within a population, quantifies the expected proportion of heterozygous genotypes under Hardy–Weinberg equilibrium [46]. It represents the probability that two randomly selected alleles from a population are different at a specific locus. The *He* is calculated as below:*He* = 1 − ∑*pi*^2^(1)
where *pi* is the frequency of the *i*-th allele in the population [47].

Individual heterozygosity of scion and rootstock accessions was assessed using the GENHET package for SSR markers and the -het command in VCFtools for SNP markers [48]. Two individual heterozygosity measures were considered [49]:

PHt: Proportion of heterozygous loci in an individual (number of heterozygous loci/number of genotyped loci).

Hs_obs: standardized heterozygosity based on the mean observed heterozygosity (PHt/mean observed heterozygosity of genotyped loci).

In addition to gene diversity, polymorphism information content (PIC) also reflects the genetic properties of SNPs within a population [50]. PIC measures the ability of markers to detect polymorphism within a population and is used to evaluate the informativeness of genetic markers for distinguishing between different alleles. PIC is calculated as:*PIC* = 1 − ∑*pi*^2^ − ∑∑2*pi*^2^*pj*^2^(2)
where *pi* and *pj* are the frequencies of the *i*-th and *j*-th alleles, respectively [51].

### 2.5. Population Structure Analysis

The population structure of the germplasm was analysed using STRUCTURE v.2.3.4 with a Bayesian clustering approach based on the admixture model with correlated allele frequencies between populations [52]. Five runs were performed for each value of K (ranging from K = 1 to K = 10) to estimate the optimal number of genetic clusters. Each simulation consisted of 20,000 burn-in and 100,000 iterations. Based on the log probability of the data [LnP(D)] and delta K (ΔK), which assesses the rate of change in [LnP(D)] between consecutive K-values, the optimal K-value was determined [53]. The STRUCTURE Q-matrix was visualized using pophelper R package [54]. The fixation index (Fst) was calculated to measure genetic differentiation between subpopulations within the entire population [55]:(3)$$\mathrm{Fst}=\frac{Ht-Hs}{Ht}$$
where *Ht* is the total expected heterozygosity of the whole population, while *Hs* represents the average expected heterozygosity across all subpopulations.

Principal component analysis (PCA) of rootstocks was conducted using PLINK 2.0 based on the standardized covariance matrix of the genetic distances between accessions [56]. A maximum-likelihood phylogenetic tree was generated using FastTree 2.1 [57] and visualized with Interactive Tree of Life (iTOL) v5 [58]. Hierarchical clustering analysis was performed to identify clusters among 94 passion fruit rootstock accessions based on a dissimilarity matrix derived from genetic distance data [59]. A dendrogram representing the clustering results of rootstock accessions was constructed using the average linkage method (UPGMA) in RStudio version 4.4.0. The weighted neighbour-joining dendrogram was constructed for 95 scion accessions based on genetic dissimilarity matrices in DARwin version 6.0.21. Principal coordinate analysis (PCoA) for scion accessions was performed with GenAlEx version 6.5 based on standardized genetic distances [60].

## 3. Results

### 3.1. Genetic Properties of SNP Markers

A total of 24,096 SNP markers were identified with PIC values varying from 0.1 (6919 SNPs) to 0.5 (4748 SNPs). These markers were found to be informative, with an average PIC value of 0.20. Approximately 29% of markers were classified within the highest PIC range (0.4 to 0.5) and only 5.1% in the range from 0.3 to 0.4, while 40.2% of markers had a PIC value less than 0.1 (Figure 1).

### 3.2. Population Structure and Genetic Diversity Analysis

The Bayesian cluster analysis performed in STRUCTURE revealed the genetic composition of 94 passion fruit rootstock accessions. The optimal number of genetic clusters (K) was determined by plotting it against ΔK, which showed a peak at K = 6 (Figure 2a). This suggests that the accessions can be classified into six distinct subpopulations (POP1–POP6), comprising 3%, 0.3%, 18.4%, 22.9%, 49.2%, and 6.2% of the total accessions, respectively (Table 1). Additionally, a secondary peak at K = 4 was also observed (Figure 2a), indicating a possible alternative clustering pattern.

The population assignment test in STRUCTURE further illustrated the membership proportions of individuals in each rootstock population (Figure 2b). The estimated membership proportion (Q) indicated that the Pandora accessions were exclusively allocated to POP5 (Appendix A). *P. edulis flavicarpa* accessions predominantly belonged to POP3, with minor contributions from POP1, POP2, and POP5 (Figure 2b). Hybrid accessions showed mixed ancestry; although some were assigned primarily to POP5, the majority of hybrid accessions displayed admixture between POP5 and either POP4 or POP3 (Figure 2b). McGuffies Red accessions were predominantly derived from POP4, with two individuals exhibiting shared ancestry with POP3. The McLeod rootstock accession displayed a broad genetic background, incorporating contributions from five subpopulations (POP1, POP2, POP3, POP4, POP5). The Heuston accessions were mainly grouped into POP1 with minimal genetic contributions from POP3. Notably, most SP16 accessions clustered within POP6, except for SP16-42D, which exhibited a different genetic background (Figure 2b).

The genetic diversity within each population was assessed by estimating the heterozygosity (*He*) (Table 1). POP2 had the highest *He* (0.386), whereas POP4 exhibited the lowest diversity (*He* = 0.013). The expected heterozygosity values for the other subpopulations were 0.063 for POP1, 0.015 for POP3, 0.028 for POP5, and 0.328 for POP6 (Table 1). Pairwise net nucleotide distance analysis indicated that POP4 exhibited substantial genetic divergence from other populations, with distance ranging from 0.158 (POP2) to 0.278 (POP5). In contrast, the smallest genetic distance was observed between POP2 and POP6 (0.030) (Table 1). The fixation index (Fst) estimated from the STRUCTURE analysis suggested a significant level of genetic divergence within subpopulations (Table 1). POP1, POP3, POP4, and POP5 exhibited high differentiation (Fst = 0.922, 0.960, 0.971, and 0.934, respectively), whereas POP6 (Fst = 0.252) displayed comparatively lower genetic divergence (Table 1).

Consistent with STRUCTURE results, PCA based on the pairwise genetic distance matrix for all 94 rootstock accessions identified six distinct clusters (Figure 2c). The first two axes of the PCA accounted for 30.9% and 22.3% of the total genetic divergence, respectively (Figure 2c). The Pandora accessions, the currently used rootstock in Australia, were positioned in the lower-left quadrant of the principal component analysis (PCA) (Figure 2c). In contrast, the hybrid accessions exhibited considerable genetic diversity, as they were widely distributed across three quadrants of the PCA. The *P. edulis flavicarpa*, Heuston, McGuffies Red, and McLeod rootstock accessions were predominantly situated in the upper-right quadrants (Figure 2c). Additionally, only six SP16 accessions were located in the upper-left quadrant of the PCA (Figure 2c).

By contrast, only two distinct groups (POP1 and POP2) were found in the STRUCTURE analysis of the 95 passion fruit scion accessions and subpopulations, representing 56.8% and 43.2% of total accessions (Figure 3). The average membership proportion of each scion accession is shown in Appendix A. The *He* of the subpopulations were 0.480 for POP1 and 0.585 for POP2 (Appendix A). Principal Coordinate Analysis (PCoA) of 95 scion accessions exhibited two clusters with the first two axes explaining 34.52% and 22.32% of the total genetic variation (Appendix A).

Individual heterozygosity estimates traditionally served as proxies for inbreeding coefficients to assess the presence of inbreeding depression. In this study, individual observed heterozygosity was estimated to identify parents that may contribute to larger genetic variation. The standardized observed heterozygosity of the 95 scion accessions ranged from 0 to 2.007, with an average of 0.996 (Appendix A). Additionally, the mean value of standardized observed heterozygosity of the 94 rootstock accessions based on SNP markers was 1, ranging from 0.131 to 3.041 (Appendix A).

### 3.3. Genetic Relationships and Hierarchical Clustering

To elucidate the relationships among the rootstock accessions, a maximum-likelihood phylogenetic tree and a UPGMA dendrogram were constructed (Figure 4 and Figure 5). Analysis of genetic relationships among the 94 passion fruit accessions revealed significant genetic variation within rootstock populations (Figure 4). The genetic dissimilarities among the cultivars assessed using DArT SNP markers varied from 0.001 to 0.695 (Appendix A). Accessions 17-009 (McGuffies Red × Griffiths Pandora) and 17_007 (McGuffies Red) exhibited the highest genetic dissimilarities with 17-018 (McGuffies Red × McLeod Pandora) (Figure 4).

Consistent with the results of principal component analysis (PCA), six major clusters were identified in both the hierarchical clustering dendrogram and phylogenetic tree of the 94 rootstock accessions (Figure 4 and Figure 5). Cluster 1 (C1) comprised 53% of the total cultivars, predominantly represented by hybrid and Pandora accessions. Of the total accessions, 6% were grouped into Cluster 2 (C2), which was exclusively occupied by SP16 accessions. Cluster 3 (C3) consisted of 13% of the total accessions and was composed solely of hybrids. Seven *P. edulis flavicarpa* cultivars, along with one hybrid and McLeod rootstock accession, were grouped into Cluster 4 (C4). In addition, three Heuston accessions (3%) formed a separate cluster (C5). Finally, eight McGuffies Red accessions and their hybrids (McGuffies Red × Griffiths Pandora; DPI *P. flavicarpa* × McGuffies Red) were clustered together in Cluster 6 (C6). For the 95 passion fruit scion accessions, a weighted neighbour-joining dendrogram was constructed using a genetic dissimilarity matrix to visualize the relationships among the accessions (Appendix A).

## 4. Discussion

Understanding genetic variation in germplasm is crucial for conservation, resource management, and utilization in breeding programs [2]. This study used DArT SNP markers to assess the genetic diversity and population structure in 94 passion fruit rootstock accessions and eight SSR markers for 95 scion accessions. The analysis revealed significant genetic diversity, particularly from hybrid rootstock accessions, which offers a foundation for developing varieties with improved resistance and adaptability.

### 4.1. DArT SNP Marker Platform in Characterization of Passiflora

The study underscores the effectiveness of the DArT SNP marker platform in evaluating genetic diversity and characterizing *Passiflora* germplasm. Previous research on passion fruit predominantly utilized microsatellite markers (SSR) for genetic diversity analysis. For instance, a PIC of 0.46 was obtained among 36 yellow and purple passion fruit accessions with 23 SSR loci [17]. A study by Araya et al. [61] observed that PIC values for 18 selected SSR markers varied from 0.46 to 0.77 with an average of 0.60 in 79 *Passiflora* species, including *P. edulis*. The polymorphism analysis of eight SSR loci conducted by Wu et al. [40] reported an average polymorphism information content (PIC) of 0.614 in the genetic assessment of 87 Passiflora germplasm resources. In our study, comparable PIC results were observed for eight SSR markers, ranging from 0.339 to 0.704 (Appendix A). However, linking SSR markers to phenotypic data has proven challenging due to their uneven genomic distribution, which results in variable trait influences depending on the proximity of each marker to relevant genes [62]. Consequently, despite their relatively lower average PIC values, the development of SNP markers is expected to provide a more robust foundation for phenotypic trait analysis in marker-assisted breeding programs [27].

In this study, a total of 24,096 DArT SNP makers were developed, providing insights into the passion fruit genome and future breeding efforts. These markers are valuable for studying genetic diversity, identification, and utilization of *Passiflora* germplasm. They provide efficient molecular tools for further exploration of genetic variation patterns in *Passiflora* germplasm resources and advancing efforts to maximize the potential of this species in agricultural, medicinal and ornamental applications.

The PIC value reflects the informativeness of SNP markers and their ability to distinguish among different genotypes [63]. According to Botstein et al. [51], the classification of markers based on PIC values applies to multi-allelic markers, whereas for biallelic markers like SNPs, the maximum possible PIC value is 0.5, which occurs when both alleles are equally frequent [38]. Hence, SNPs with PIC values 0.25–0.5 are considered to be informative markers for genetic studies. In this study, 6986 SNP markers were identified as highly informative (PIC = 0.4–0.5), with an average of 0.460 (Figure 1). Comparable findings were reported in a previous study by Anderson et al. [27], which characterized 59 *Passiflora* species in Florida. The study found an average PIC value of 0.187 across 26,191 SNP loci, with values ranging from 0.091 to 0.375. For 5817 *P. edulis* SNPs, the average PIC value was slightly higher at 0.241 [27]. Another published study by Rodriguez Castillo et al. [28] investigated 88 accessions of purple passion fruit accessions, utilizing 966 informative SNPs with an average PIC of 0.420. Although this study reported high PIC values, the number of SNPs was significantly lower than in our study, which included 24,096 SNPs.

### 4.2. Analysis of Population Structure and Genetic Diversity

Population structure analysis enhances the understanding of genetic diversity and is essential for accurate association mapping by identifying subpopulations, thereby controlling for genetic similarities that could lead to false positives in trait associations [64]. Therefore, assessing the underlying population structure is a crucial initial step in conducting genome-wide association studies (GWAS) to accurately identify true associations between markers and traits and the underlying genes that regulate these traits [50]. In this context, our study revealed that both STRUCTURE analysis, which determined an optimal K value of 6 (Figure 2), and principal component analysis (PCA) (Figure 2) indicated that the 94 passion fruit rootstock accessions could be classified into six distinct subgroups. However, the structure of the passion fruit rootstock accessions observed in our study is not consistent with the findings of other research investigating the population structure of *P. edulis* accessions. For example, Rodriguez Castillo et al. [28] found that the most probable number of subpopulations of 88 purple passion fruit genotypes and five yellow varieties based on 3820 informative SNP loci was determined to be K = 2, whereas K = 3 for the subset comprising only purple passion fruit. This discrepancy in the number of subpopulations can be attributed to several factors, including differences in population composition, variations in analytical approaches and the specific SNP subsets used [27], as well as the diverse origins of the passion fruit accessions in our study. However, only two subpopulations were found in our 95 scion accessions with eight SSR markers, aligning with the findings of Rodriguez Castillo et al. [28]. This suggests a relatively higher level of complexity within the analysed passion fruit rootstock populations, providing a valuable resource for breeding programs. This diversity supports the strategic application of selection pressure to improve key traits, such as disease resistance, stress tolerance, and productivity in the development of enhanced passion fruit varieties [27].

*He* estimates the average level of heterozygosity and genetic differentiation among individuals within a population [65]. We found a mean *He* value of 0.533 across two scion subpopulations with eight SSR loci (Appendix A). By contrast, an average *He* of only 0.139 was observed for the 94 rootstock accessions based on SNP markers (Table 1). Previous studies analysing genetic diversity in passion fruit through molecular markers have revealed varying degrees of genetic variability and distinct patterns of genetic relationships within the species. For example, an analysis of genetic diversity in 87 passion fruit germplasm accessions in China, utilizing single sequence repeat (SSR) fluorescent markers, yielded an average expected heterozygosity of 0.202 [40]. Another study on the genetic variability and structure of 51 Colombian commercial yellow passion fruit accessions reported an average *He* of 0.78, based on analysis with six microsatellite markers [21]. In contrast, the findings of Ortiz et al. [66] indicated significant genetic homogeneity among purple passion fruit cultivars in Colombia based on analyses using microsatellite and AFLP markers. Additionally, an average expected heterozygosity of 0.47 was reported by Rodriguez Castillo et al. [28] across 88 purple passion fruit accessions with 966 SNPs. There are several factors that may influence genetic variability within a population, including seed dispersal, gene flow, natural selection, geographic distribution, and the centre of diversity [67]. In addition, the selection and combination of different markers may influence the identification of passion fruit cultivars, potentially affecting the results [68]. Overall, discrepancies observed between our findings and studies utilizing alternative molecular markers may be attributed to variations in marker selection, the specific accessions incorporated, and the methods applied for calculating heterozygosity [27].

### 4.3. Implications on Breeding Efforts

Genetic variability among these *P. edulis* accessions enhances the understanding of their genetic basis, providing promising opportunities for breeding applications. Australian passion fruit rootstocks primarily depend on Pandora accessions, including Griffiths Pandora, Bunnings Pandora, and McLeod Pandora. These accessions in our STRUCTURE analysis were exclusively clustered in Subpopulation 5, indicating high genetic similarity among them (Figure 2b). However, current passion fruit production faces challenges due to escalating biotic and abiotic stresses [69]. In contrast, hybrids derived from these Pandora accessions in our study offer valuable resources for enhancing the genetic diversity of Australian rootstock germplasm. The high heterozygosity observed in these hybrid rootstock accessions offers significant advantages in variable environments, promoting optimal plant growth and enhanced overall fitness [70]. Therefore, the genetic data obtained and the characterized accessions in our study will provide a robust foundation for future efforts in Australian passion fruit breeding programs to expand the germplasm and improve resistance to Fusarium wilt, woodiness virus disease, and cold tolerance.

Individual heterozygosity measures can provide insights into levels of outcrossing and genetic variation among potential parents. Higher heterozygosity in parents may indicate greater genetic diversity, which can be beneficial for generating offspring with broader genetic variation. In this study, highly heterozygous scion and rootstock accessions were identified based on the proportion of heterozygous loci in an individual and standardized heterozygosity (Appendix A). However, other factors like allele diversity, genetic distances and compatibility between parents also play crucial roles in determining the genetic variation of progeny [71]. Therefore, it is recommended to combine heterozygosity with genetic distance and population structure information to select parents for maximizing genetic variation and hybrid vigour.

The majority of hybrid accessions (68%) in our study exhibited mixed genetic compositions derived from multiple subpopulations, while a subset of hybrids (32%) predominantly reflected genetic contributions from a single subpopulation (Figure 2). This pattern may be attributed to self-pollination and highlights the potential utility of the selected genetic markers from this study to verify the origin of planting materials, ensuring they result from the intended cross-hybridization rather than self-fertilization. However, self-pollination in passion fruit may lead to a reduction in genetic diversity, particularly in cultivars selected for commercial production [27]. Passion fruit exhibits an open flower structure, characterized by the position of the style above the stamens and the abundance of anthers and pollen, which facilitates manual cross-pollination [27]. These attributes underscore the potential for utilizing controlled crosses or leveraging self-incompatibility to exploit heterosis and maintain genetic diversity, ensuring sustainable passion fruit production. Although this study successfully generated 36 passion fruit rootstock hybrids, further research is necessary to identify optimal hybrid combinations that maximize fruit production and enhance tolerance to both biotic and abiotic stresses. Furthermore, the compatibility and interactions between rootstock and scion accessions in this study require further analysis to better understand their influence on overall plant performance and productivity.

## 5. Conclusions

This study provides critical insights into the genetic diversity of Australian passion fruit germplasm, highlighting the substantial variation present within rootstock populations. The utilization of high-density DArT SNP markers enabled a comprehensive analysis of genetic relationships among 94 rootstock accessions, revealing six distinct genetic groups that underscore the potential for targeted breeding strategies. The allelic diversity identified offers a valuable resource for enhancing rootstock traits, which is essential for improving productivity, fruit quality, and resilience in Australian passion fruit breeding programs. However, only two genetic groups were found for 95 scion accessions. Therefore, our findings emphasize the necessity of preserving and utilizing rootstock genetic diversity to develop superior cultivars capable of thriving in diverse environmental conditions. In future research, the core accessions identified in this diversity study will be integrated into Australian rootstock breeding programs for further physiological and agronomical evaluations.

## Figures and Tables

**Figure 1 biotech-14-00037-f001:**
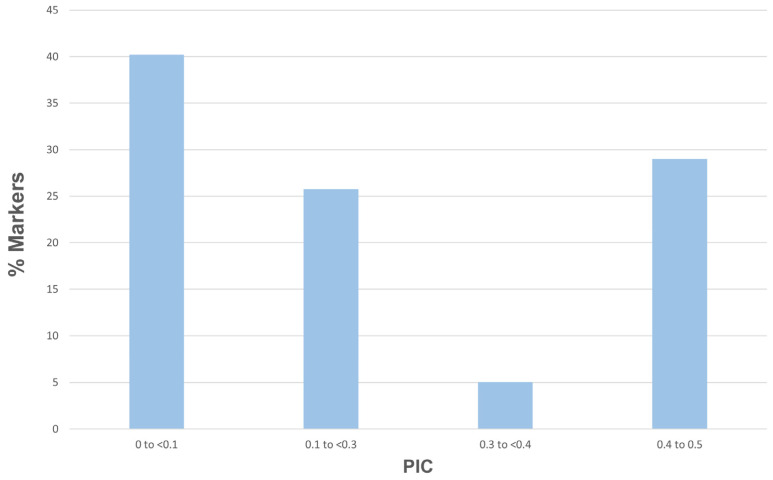
Distribution of PIC values for SNP markers used in the study.

**Figure 2 biotech-14-00037-f002:**
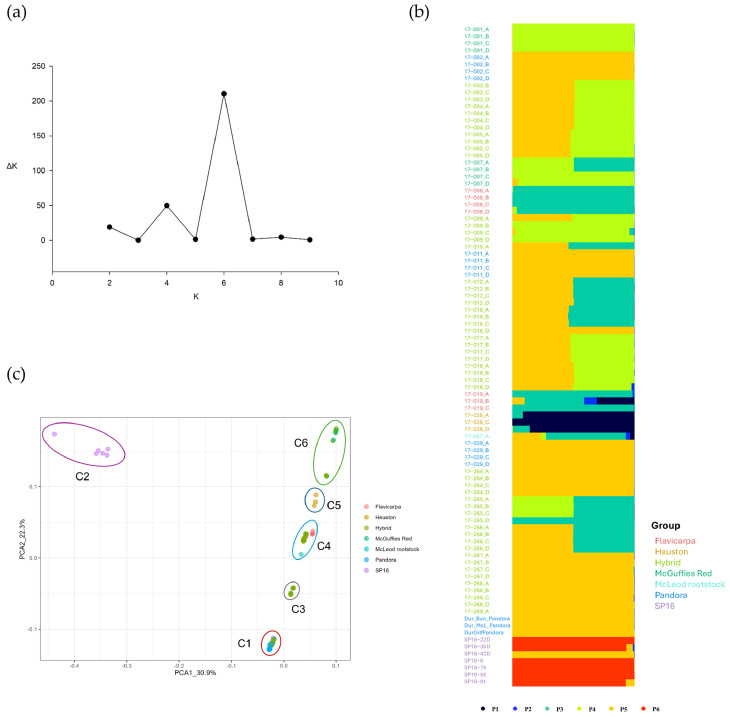
The population structure of the 94 passion fruit rootstock accessions. (**a**) Plot of K against ΔK to determine the optimum K value for the STRUCTURE analysis of 94 rootstock accessions; (**b**) the proportion of individuals assigned to six population groups; (**c**) principal component analysis (PCA) based on genetic distance across 94 rootstock accessions.

**Figure 3 biotech-14-00037-f003:**
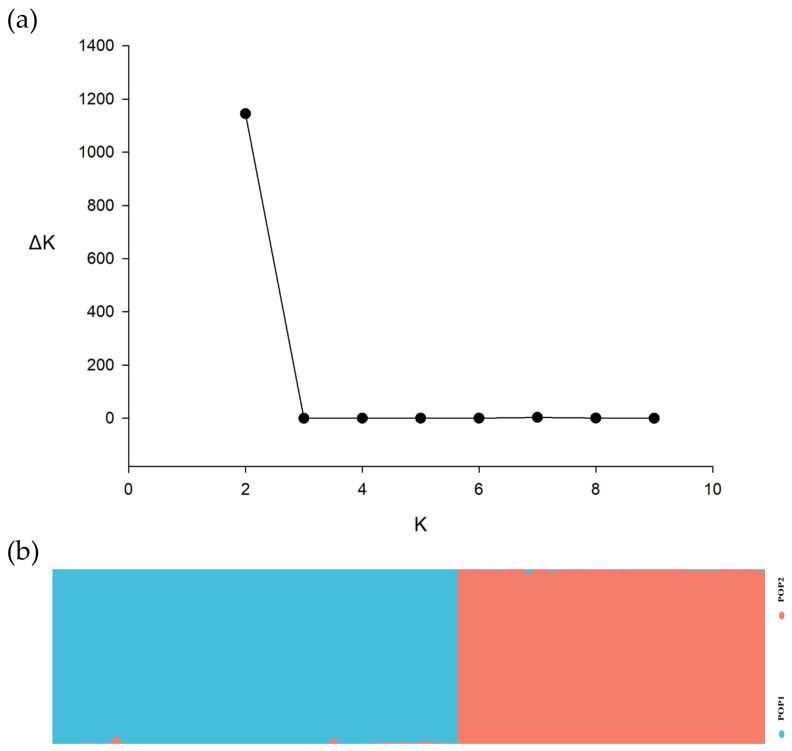
The population structure of the 95 passion fruit scion accessions. (**a**) Plot of K against ΔK to determine the optimum K value for the STRUCTURE analysis of 95 scion accessions; (**b**) proportion of assignment of individuals to two population groups.

**Figure 4 biotech-14-00037-f004:**
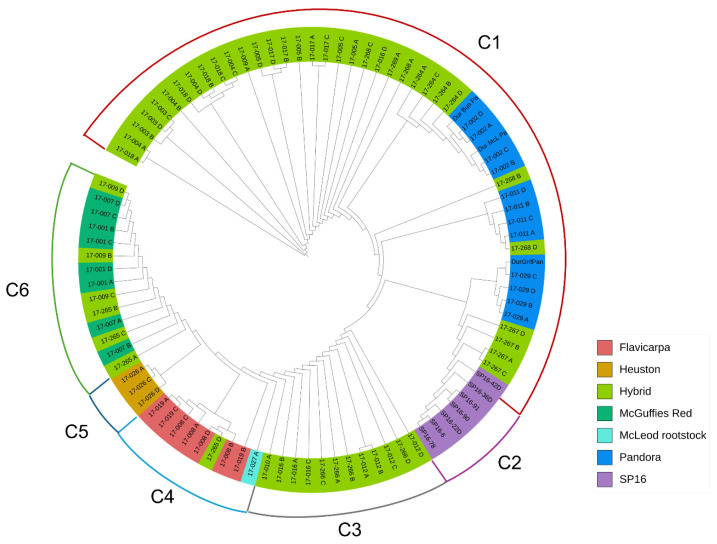
The phylogenetic relationships among the 94 rootstock accessions. The tree was categorized into six clusters depending on STRUCTURE subpopulations for K = 6.

**Figure 5 biotech-14-00037-f005:**
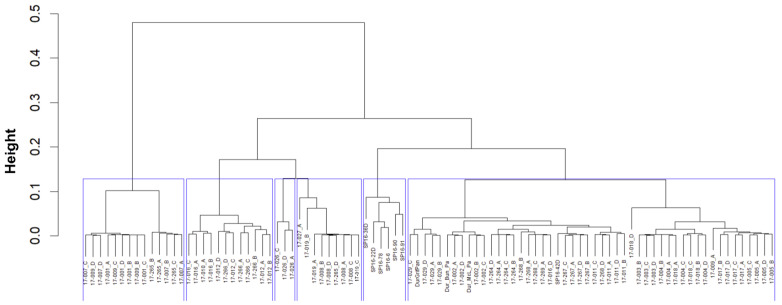
The dendrogram showing the hierarchical clustering results of the 94 rootstock accessions from average linkage clustering. Six clusters are denoted by blue lines.

**Table 1 biotech-14-00037-t001:** The STRUCTURE results of 94 rootstock accessions based on DArT SNP markers for genetic divergence between (net nucleotide distance) and within populations (expected heterozygosity), the fixation index (Fst), and the proportion of membership of the populations.

POP	Net Nucleotide Distance					Exp. Het	Mean Fst	Prop. Mem
	POP2	POP3	POP4	POP5	POP6			
POP1	0.124	0.141	0.241	0.245	0.224	0.063	0.922	0.030
POP2		0.121	0.158	0.095	0.030	0.386	0.014	0.003
POP3			0.241	0.205	0.217	0.015	0.960	0.184
POP4				0.278	0.263	0.013	0.971	0.229
POP5					0.123	0.028	0.934	0.492
POP6						0.328	0.252	0.062

## Data Availability

The data supporting this study is available from the corresponding author upon reasonable request.

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
