# Peer review of "Exploring Genetic Diversity and Population Structure of Australian Passion Fruit Germplasm"

_biotech, 2025, doi:10.3390/biotech14020037_

Round 1

Reviewer 1 Report

Comments and Suggestions for Authors

The manuscript is well written with the proper introduction to the problem, setting of hypothesis, planning of experiment, standard methodology, robust statistical analysis, results presented with illustrations, and a conclusion.

The manuscript can be ACCEPTED for publication after MINOR REVISIONS

(1) Use uniformly 'passionfruit' or 'passionfruit' throughout the manuscript

(2) Sections 4.2 and 4.3, the discussion needs to be shortened as there a too much content in it.

Author Response

Comments 1: Use uniformly 'passion fruit' or 'passionfruit' throughout the manuscript.

Response 1: Thank you for pointing it out. We have uniformly used ‘passion fruit’ throughout the manuscript.

Comments 2: Sections 4.2 and 4.3, the discussion needs to be shortened as there a too much content in it.

Response 2: Thanks for your suggestion. We have deleted some redundant sentences, such as “Importantly, the PCA results aligned with those obtained from the STRUCTURE analysis.” in Line 352 and “using heterozygosity alone may not fully capture the potential for generating large genetic variation.” in Line 410.

Reviewer 2 Report

Comments and Suggestions for Authors

The manuscript evaluated germplasm diversity for Australian passionfruit breeding programs. The genetic diversity of Australian passionfruit, including 94 rootstocks and 95 scions, was evaluated. Rootstocks were genotyped using high-density 24k Diversity Arrays Technology (DArT)-based single nucleotide polymorphism (SNP) markers, and genetic characterization of scions was conducted using eight simple sequence repeat (SSR) markers. Bayesian cluster analysis showed that the rootstock population was divided into six distinct genetic groups, whereas only two subpopulations were identified among the scion accessions. SNP-based genotyping further highlighted the allelic diversity of Australian rootstocks, suggesting a rich reservoir of genetic traits for rootstock improvement. In general, the manuscript is well-written and the experiments were well-performed.

1) Why were rootstocks genotyped by using SNP markers, while scions were genotyped by using SSR markers? The reasons should be indicated.

2) All abbreviations should be introduced with full names, such as ISSR and RAPD in Line 77.

3) Most numbers and letters in the figures are too small to be seen clearly. Larger figures should be provided. Alternatively, all numbers and letters should be enlarged.

4) geographical distribution of different genotypes should be shown.

5) Some references are incomplete (press names or page numbers are missing), such as Ref. 9, Ref. 19 and Ref. 26. The pages numbers of Ref. 34 should be replaced by the manuscript ID.

Author Response

Comments 1: Why were rootstocks genotyped by using SNP markers, while scions were genotyped by using SSR markers? The reasons should be indicated.

Response 1: Thank you for your valuable comment. The decision to use SNP markers for rootstocks and SSR markers for scions was based on a combination of resource availability and the specific goals of each germplasm analysis. Although rootstocks play an important role in passion fruit production, there is limited study about genetic characterization on passion fruit rootstocks to support breeding programs. Therefore, high-density SNP genotyping was used for rootstocks to enable detailed analysis of allelic diversity and population structure for further physiological and agronomical evaluations. In contrast, the scion accessions had previously been characterized using SSR markers as part of an earlier evaluation, and this dataset was used to complement the current study. Additionally, SSR markers, while fewer in number, provided sufficient resolution to differentiate among the scion accessions for the purposes of this study.

Comments 2: All abbreviations should be introduced with full names, such as ISSR and RAPD in Line 77.

Response 2: Agree. We have revised in Line 77-78.

Comments 3: Most numbers and letters in the figures are too small to be seen clearly. Larger figures should be provided. Alternatively, all numbers and letters should be enlarged.

Response 3: Thank you for the suggestions. We have provided larger figures in the revised manuscript.

Comments 4: Geographical distribution of different genotypes should be shown.

Response 4: We appreciate the suggestion regarding the geographic distribution of the genotypes. However, in the context of this study, the passionfruit accessions analysed are cultivated varieties rather than wild populations. These genotypes have been propagated and redistributed across regions over time through seed or vegetative material, making their original geographic origins difficult to verify with certainty. For instance, while some genotypes such as ‘McGuffies Red’ and ‘Pandora’ were collected in Queensland, the exact provenance of many genotypes is unclear due to a lack of historical documentation and extensive movement within breeding programs. Therefore, it may not be feasible to obtain an accurate geographical distribution map of different genotypes, and such data may not reflect the biological or breeding relevance of the materials used.

Comments 5: Some references are incomplete (press names or page numbers are missing), such as Ref. 9, Ref. 19 and Ref. 26. The pages numbers of Ref. 34 should be replaced by the manuscript ID.

Response 5: Thank you so much and we have revised them.

Reviewer 3 Report

Comments and Suggestions for Authors

A brief summary

Genetic variability and population structure are important components when conducting evaluation of cultivars in order to enhance their performances in breeding. The authors of this manuscript conducted the study in this area on Australian passionfruit, a highly-values tropical fruit. The manuscript is organized nicely and well-written. The introduction provided a good background and overview leading to why this study was conducted. The materials and methods detailed the methodology on how the experiments were done and could be replicated by others who would like to do similar studies. The figures presented in the results section enhanced the explanation in results and discussion sections. The results from this study are beneficial for other scientists who plan to conduct similar research efforts. In the discussion section the authors presented potential future use of the results of this study and the next step to be researched.

Specific comments:

Below are several suggestions to improve the quality of the manuscript.

Some of the suggested areas are because the style of writing is not common for peer-reviewed paper. I am referring to those similar to example in line 118 that I pointed below. I will not list them one by one, but authors need to find them within the manuscript. It should list first author cited references then followed by citation number: …. method by [##], described by [##] etc. should be … method by ‘author name’ [##], described by ‘author name’ [##] etc.

Introduction (page 2-3)

Lines 77-78

Rearrange the phrase to this: … methods, including the following markers: SSR [13, 18-21] …, and SNP [27, 28].

Line 92

Delete ‘therefore’.

Line 86

You indicated that ‘DArT markers have been extensively employed across various crops …’ but only provided 3 references. Provide 4-5 more references that showed the extensive use of these markers.

Materials and Methods (page 2)

Lines 101-107

When were the samples collected?

Why 94 rootstocks and 95 scions collected? In other words, why did you choose these sample numbers? Please explain.

Line 109

From where did you obtain the fresh samples? Please state the location.

Line 111

Delete ‘described by’, or state it this way: … protocol described by Doyle [37].

Line 113

List the manufacturer of the spectrophotometer in parentheses.

Lines 109-114

How do you collect the leaves? Do you put them in tubes or containers? How do you rapidly frozen? There are not enough details in this section, please revise.

Line 116

Spell out 94; numbers at the beginning of a sentence must be spelled out.

Line 118

Delete ‘by’ and replaced with ‘previously described’. Or, revise it this way: … as described by Barilli et al. [39].

Results (page 6)

Line 214

… by estimating the He … -- change to: … by estimating the heterozygosity (He) …

Discussion (page 9-10)

Lines 308, 324, 350, 358, 373, 376

See my previous comments

References (page 13-14)

Line 483

… (accessed on 21 March) -- what year???

Line 515-516

Reference is incomplete. Please revise.

Author Response

Comments 1: Some of the suggested areas are because the style of writing is not common for peer-reviewed paper. I am referring to those similar to example in line 118 that I pointed below. I will not list them one by one, but authors need to find them within the manuscript. It should list first author cited references then followed by citation number: …. method by [##], described by [##] etc. should be … method by ‘author name’ [##], described by ‘author name’ [##] etc.

Response 1: Thank you for pointing it out. We have revised it accordingly.

Comments 2: Lines 77-78 Rearrange the phrase to this: … methods, including the following markers: SSR [13, 18-21] …, and SNP [27, 28].

Response 2: Rearranged in Line 77.

Comments 3: Line 92 Delete ‘therefore’.

Response 3: Deleted in Line 94.

Comments 4: Line 86 You indicated that ‘DArT markers have been extensively employed across various crops …’ but only provided 3 references. Provide 4-5 more references that showed the extensive use of these markers.

Response 4: Thanks. We have provided 4 more references in Line 89-90.

Comments 5: Lines 101-107 When were the samples collected?

Response 5: The samples were collected in 2018, and we have added it in Line 112.

Comments 6: Why 94 rootstocks and 95 scions collected? In other words, why did you choose these sample numbers? Please explain.

Response 6: Thank you for your insightful comment. The selection of 94 rootstock and 95 scion accessions was based on the availability of genetically and agronomically relevant material from the Australian passionfruit germplasm collections. These accessions represent the core diversity maintained and used in Australian passionfruit breeding programs.

Comments 7: Line 109 From where did you obtain the fresh samples? Please state the location.

 Response 7: The leaf samples used in this study were collected from mature vines growing at the Duranbah trial site, located in northern New South Wales, Australia. This information has been added in Line 113 for clarity.

Comments 8: Line 111 Delete ‘described by’, or state it this way: … protocol described by Doyle [37].

Response 8: Revised in Line 117.

Comments 9: Line 113 List the manufacturer of the spectrophotometer in parentheses.

Response 9: Listed in Line 119.

Comments 10: Lines 109-114 How do you collect the leaves? Do you put them in tubes or containers? How do you rapidly frozen? There are not enough details in this section, please revise.

Response 10: Thanks for pointing it out. We have provided more information in Line 113-115.

Comments 11: Line 116 Spell out 94; numbers at the beginning of a sentence must be spelled out.

Response 11: Revised in Line 122 and 127.

Comments 12: Line 118 Delete ‘by’ and replaced with ‘previously described’. Or, revise it this way: … as described by Barilli et al. [39].

Response 12: We have revised it in Line 124-125.

Comments 13: Line 214 … by estimating the He … -- change to: … by estimating the heterozygosity (He) …

Response 13: Revised in Line 220. 

Comments 14: Lines 308, 324, 350, 358, 373, 376 See my previous comments

Response 14: Thanks for pointing it out. We have revised accordingly.

Comments 15: Line 483 … (accessed on 21 March) -- what year???

Response 15: Thank you and we have revised it in Line 483 and 486.

Comments 16: Line 515-516 Reference is incomplete. Please revise.

Response 16: We have revised it in Line 520.